# Decoding the Role of Insulin-like Growth Factor 1 and Its Isoforms in Breast Cancer

**DOI:** 10.3390/ijms25179302

**Published:** 2024-08-27

**Authors:** Amalia Kotsifaki, Sousanna Maroulaki, Efthymios Karalexis, Martha Stathaki, Athanasios Armakolas

**Affiliations:** 1Physiology Laboratory, Medical School, National and Kapodistrian University of Athens, 11527 Athens, Greece; amkotsifaki@med.uoa.gr (A.K.); souzannamaroulaki@gmail.com (S.M.); euthimiskaralexis12@gmail.com (E.K.); 2Surgical Clinic, “Elena Venizelou” General Hospital, 11521 Athens, Greece; stathakimg@yahoo.gr

**Keywords:** insulin-like growth factor-1 (IGF-1), IGF-1 receptor (IGF-1R), IGF-1 isoforms, breast cancer (BC), tumorigenesis, apoptosis inhibition, signal transduction, therapeutic targets, cancer metasta-sis, novel anti-cancer agents, oncogenic signaling pathways

## Abstract

Insulin-like Growth Factor-1 (IGF-1) is a crucial mitogenic factor with important functions in the mammary gland, mainly through its interaction with the IGF-1 receptor (IGF-1R). This interaction activates a complex signaling network that promotes cell proliferation, epithelial to mesenchymal transition (EMT) and inhibits apoptosis. Despite extensive research, the precise molecular pathways and intracellular mechanisms activated by IGF-1, in cancer, remain poorly understood. Recent evidence highlights the essential roles of IGF-1 and its isoforms in breast cancer (BC) development, progression, and metastasis. The peptides that define the IGF-1 isoforms—IGF-1Ea, IGF-1Eb, and IGF-1Ec—act as key points of convergence for various signaling pathways that influence the growth, metastasis and survival of BC cells. The aim of this review is to provide a detailed exami-nation of the role of the mature IGF-1 and its isoforms in BC biology and their potential use as possible therapeutical targets.

## 1. Introduction

Breast cancer (BC) is a complex and prevalent disease, ranking as the second leading cause of cancer-related mortality worldwide and the most common malignancy among women [1]. BC represents 23% of all cancer cases and 14% of cancer-related deaths, with its incidence surpassing that of lung cancer globally since 2020 [2]. The disease imposes a significant emotional, physical, and financial burden on patients, families, and healthcare systems [3].

BC is classified into four main subtypes based on molecular characteristics and hormone receptor (HR) status assessed through immunohistochemistry (IHC): Luminal A (ER+, PR+, Ki67 <20%), Luminal B (ER+, PR+/−, HER2+/−, Ki67 >20%), HER2-overexpressing (HER2+), and Triple-negative BC (TNBC), which lacks expression of ER, PR, and HER2 [4]. These subtypes differ significantly in their biological characteristics, clinical behavior, and response to treatment, highlighting the need for individualized treatment strategies [5].

In recent years, the role of insulin-like growth factor 1 (IGF-1) in BC has gained considerable attention due to its strong mitogenic and anti-apoptotic effects, which are crucial for cell growth and survival [6]. The IGF-1 system, IGF-binding proteins (IGFBPs), and the IGF-1 receptor (IGF-1R), is a critical component in human physiology and plays a pivotal role in the development and function of various tissues, including the mammary gland [7]. Various isoforms of IGF-1, including IGF-Ea, IGF-Eb, and IGF-Ec, have been identified, each potentially playing distinct roles in physiological and pathological contexts [8].

Epidemiological research indicates that elevated levels of IGF-1 in the bloodstream correlate with heightened susceptibility to prostate cancer and BC [9]. Over the past two decades, a growing body of research has implicated the IGF-1 system in the development and progression of multiple malignancies, including BC. Aberrant expression of IGF-1 system components has been observed in BCs, suggesting a crucial role in tumorigenesis [2]. Importantly, the relevance of IGF-1 signaling varies across subtypes of BC, influencing their growth and response to therapies. The IGF-1R is activated in at least 50% of BCs, with its natural ligands, IGF-1 and IGF-2, contributing to tumor growth and survival [10]. This activation promotes cellular proliferation, motility, and resistance to apoptosis, making the IGF-1 system a significant factor in cancer biology [9].

Furthermore, IGFBPs, which are widely expressed in BC, modulate interactions between IGF-1 and IGF-1R and facilitate the transport of IGF-1 and IGF-2 in extracellular fluids [11]. These proteins are linked to BC outcomes and have regulatory roles in IGF-1 signaling. The availability of IGF-1 and IGF-2 to tumor cells from both endocrine sources and autocrine/paracrine production further underscores the importance of this pathway in cancer progression [12]. In addition to IGF-1R, the insulin receptor (IR) also plays a role in the IGF signaling system, particularly its fetal A isoform. There is significant functional similarity between IGF-1R and IR, with a high degree of conservation in their kinase domains [9]. This similarity is evidenced by the shared physiological functions of the receptors, such as in tumor-associated hypoglycemia caused by elevated levels of insulin or IGF-2 from tumor cells. Both receptors activate insulin receptor substrate proteins (IRS-1 and IRS-2), which act as scaffolds for other signaling pathways, such as PI3K/AKT/mTOR and Ras/Raf/MAPK, crucial for BC subtypes [13,14].

The IGF-1 pathway has also been implicated in resistance to cancer therapies. It appears to modulate the expression of epidermal growth factor receptors (EGFR) in cancer cells, potentially inducing resistance to endocrine therapies in estrogen receptor (ER)+ BC patients [10]. Moreover, interactions between IGF-1R and the HER2 pathway may contribute to resistance against anti-HER2 therapies [2]. Additionally, components of the IGF axis interact with various genetic and environmental factors implicated in the etiology of BC, including high-penetrance genes such as BRCA1, BRCA2, p53, and PTEN [15]. 

In recent years, the IGF axis has garnered attention as a promising therapeutic target, with considerable efforts underway to translate experimental and preclinical research into effective medical protocols [16]. Overall, the IGF-1 system’s intricate involvement in cellular proliferation, survival, and resistance mechanisms highlights its importance in cancer biology and underscores the need for continued research to develop targeted therapies for BC treatment [17]. Dysregulation of the IGF-1 signaling pathway, facilitated by its isoforms IGF-1Ea, IGF-1Eb, and IGF-1Ec, has been implicated in promoting oncogenic phenotypes and contributing to therapeutic resistance in BC [7,18]. Despite advances in treatment strategies, understanding the intricate biological mechanisms underpinning BC remains crucial for improving clinical outcomes. While several biomarkers and molecular targets have been identified, only a select few have translated into routine clinical practice.

Epidemiological studies consistently link elevated levels of circulating IGF-1 with an increased susceptibility to BC, underscoring the clinical relevance of this pathway in disease progression [2,10]. Understanding the specific roles and regulatory mechanisms of IGF-1 isoforms in BC initiation and progression is essential. This review is specifically designed to address the following research questions: How do the different IGF-1 isoforms contribute to BC initiation, progression, and metastasis? What are the molecular structures and functional impacts of IGF-1Ea, IGF-1Eb, and IGF-1Ec in both normal and cancerous tissues? How can targeted therapies aimed at these isoforms overcome treatment resistance and improve patient outcomes?

To ensure a comprehensive understanding of the IGF-1 system and its role in BC, this review includes an extensive bibliographic analysis of studies published over the past 20 years. This time frame encompasses significant advancements in the field, capturing the latest insights into IGF-1 signaling mechanisms and their implications in cancer biology. The literature search was conducted across multiple reputable databases, including PubMed, Scopus, Web of Science and Embase. Keywords utilized in the search encompassed “Insulin-like Growth Factor-1 (IGF-1)”, “IGF-1 receptor (IGF-1R)”, “IGF-1 isoforms”, “breast cancer”, “tumorigenesis”, “apoptosis inhibition”, “signal transduction”, “cancer metastasis” and “therapeutic targets”. These keywords were carefully selected to cover a broad spectrum of relevant research areas, including molecular mechanisms, clinical studies, and therapeutic strategies related to IGF-1 and its isoforms in BC.

This review addresses these critical gaps by providing a focused analysis of the molecular structures, expression patterns, and functional impacts of IGF-1 isoforms in both normal and cancerous breast tissue. The investigation is especially timely as it consolidates recent findings to clarify how these isoforms influence BC initiation, progression, and metastatic behavior. Furthermore, it explores how dysregulated IGF-1 signaling pathways contribute to therapeutic resistance, a pressing issue in the treatment of BC. Decoding the intricate biological mechanisms underpinning BC remains crucial for improving clinical outcomes. Understanding the specific roles and regulatory mechanisms of IGF-1 isoforms in BC initiation and progression is essential. This review aims to comprehensively analyze the impact of three newly identified peptides arising from the IGF-1 isoforms IGF-1Ea, IGF-1Eb, and IGF-1Ec, in both normal and carcinogenic contexts, and their use as possible therapeutical targets in BC that can complement the action of the newly developed anti-IGF-1R treatment. By synthesizing the current literature and research findings, this study seeks to elucidate how dysregulated IGF-1 signaling pathways contribute to the biological processes of BC, including initiation, progression, and metastasis. Moreover, the exploration of targeted therapies aimed at specific IGF-1 isoforms offers promising avenues to overcome treatment resistance and improve clinical outcomes for BC patients.

## 2. Molecular Characteristics and Functional Roles of IGF-1 Isoforms in Health and Disease

The IGFs, which are part of the insulin-related peptide family, are integral to the IGF pathway. This pathway includes IGF (IGF-1 and IGF-2), insulin-like growth factor binding proteins (IGFBPs), and insulin-like growth factor receptors (IGF-1R and IGF-2R) [19]. To better comprehend the structure of IGFs, a comparison to insulin is helpful, as this structural similarity explains IGF-1′s ability to bind (with low affinity) to the insulin receptor [20]. IGFs are monomeric proteins with a small hydrophobic core formed by three helices, stabilized by disulfide bonds between Cys 6 and Cys 48, Cys 18 and Cys 61, and Cys 47 and Cys 52 [21]. The loop between the A and B regions is referred to as the C region, similar to the insulin molecule. Notably, the C region of IGF-1 is crucial for IGFR specificity. The IGF-C region differs from insulin at its C-termini with short extensions of eight and six residues, known as the “D region.” The C peptide region consists of 12 amino acids [22].

Unlike insulin, which functions systemically, IGFs engage in both systemic and paracrine/autocrine signaling pathways. IGF-1 is composed of 70 amino acids and has a molecular weight of 7649 Da. Similar to insulin, IGF-1 features an A and B chain linked by disulfide bonds [6]. The majority of serum IGF-1 is generated by the liver in response to growth hormone, yet liver-derived IGF-1 has been shown to be non-essential for postnatal body growth in mice [23]. Instead, locally produced non-hepatic IGF-1, acting through paracrine/autocrine mechanisms, appears to drive most of the postnatal growth-promoting effects of IGF-1 [24]. Reflective of its growth-promoting role, IGF-1 is a potent mitogen that influences various cellular functions, including cell cycle progression, apoptosis, and cellular differentiation [25].

Different IGF-1 isoforms are expressed in normal epithelial cells both in vivo and in vitro. All three IGF-1 mRNA isoforms (Ea, Eb, Ec) are found in cells in vivo, such as colonocytes, normal urothelium, and epithelial cells of the uterine cervix, as well as in in vitro cell lines such as human embryonic kidney (HEK293) and human lens epithelium (HLE-B3) cells [22,26]. Specifically, IGF1Eb isoform and class II transcripts are predominantly expressed in normal colon mucosal cells, while other studies show increased IGF-1Ea isoform and class I transcripts in normal colon tissues (Table 1) [22].

Various IGF-1 isoforms are produced through mRNA alternative splicing. The human IGF-1 gene, found on the long arm of chromosome 12, contains six exons and five long introns and is regulated by two promoters (P1 and P2) [19]. Alternative promoter usage and splicing results in six different mRNA variants: IA, IB, IC, IIA, IIB, and IIC [22]. A mature 70 kDa IGF-1 protein is encoded only by exons 3 and 4, while exons 5 and 6 are alternatively spliced to produce three C-terminal E peptides: Ea (exon 6), Eb (exon 5), and Ec (fragments of exons 5 and 6) [22,27]. The most abundant transcript is IGF1Ea, followed by IGF1Eb and IGF1Ec. These isoforms are broadly classified into two categories: Type 1 and Type 2, with Type 1 IGF-1 isoforms using exon 1 as the leader exon and Type 2 isoforms utilizing sites on exon 2 [28].

Specifically, the IGF-1Ea transcript arises from the splicing pattern of exon 1 or 2–3–4–6 of the igf-1 gene and is the primary IGF-1 mRNA produced in the liver [29]. The IGF-1Eb transcript is a splice variant of exon 1 or 2–3–4–5, while the IGF-1Ec transcript results from splicing exon 1 or 2–3–4–5–6. The presence of these different IGF-1 transcripts suggests tissue-specific auto- and/or paracrine action, as well as separate regulation of both gene promoters [29,30]. The various IGF-1 mRNA transcripts encode precursor proteins that differ in the length of their amino-terminal peptide (signal peptide) and the structure of peptide E (extension peptide or E domain) at their final end [29]. The last four amino acids in the C region seem to be responsible for the strong affinity to the IGF-1R receptor and the transmission of its signal [31]. These transcripts are generated through a combination of alternative promoter usage, alternative splicing, and different polyadenylation signals [32].

Furthermore, IGFs play a significant role in the pathogenesis of various diseases such as cancer, metabolic disorders, and cardiovascular diseases by influencing the aging process. They are also crucial for bone acquisition and maintenance [33]. Notably, IGF-1 binds to the insulin receptor IR, albeit with a lower affinity than insulin itself. This binding contributes to the modulation of glucose transport in fat and muscle tissues, inhibits glucose output from the liver, regulates hepatic glucose production, and lowers blood glucose levels while suppressing insulin production [14].

In the context of skeletal muscles, autocrine and paracrine IGF-1 are vital for muscle regeneration. Studies highlight IGF-1′s involvement in muscle growth regulation via its metabolic and anabolic effects within the growth hormone (GH)/IGF-1 axis [34]. Additionally, IGF-1 plays a crucial role in the mitogenic and myogenic processes that are essential for muscle development, regeneration, and hypertrophy. Regarding adipose tissue, in vitro studies using human mesenchymal stem cells (HMSCs) have shown that low concentrations of IGF-1 directly influence preadipocyte differentiation, clonal expansion, lipid droplet formation, and growth [8,35]. Furthermore, IGF-1 indirectly mediates lipid synthesis and lipolysis by modulating the lipolytic activities of GH and insulin [36].

The functional significance of these IGF-1 splice variants remains an area of active research, as emerging studies suggest that these isoforms may have distinct roles in various tissues and developmental stages, potentially influencing specific cellular pathways and disease processes, including cancer progression [25]. The differential expression of these isoforms in various tissues and their distinct signaling mechanisms could provide insights into their specific roles in oncogenesis, tumor progression, and metastasis [37].

## 3. The Role of IGF-1, Its Isoforms, and IGF-1R in Breast Cancer

### 3.1. IGF-1 in Breast Cancer

IGF-1 emerges as a key orchestrator in the delicate symphony of breast development, function, and BC progression [10]. This powerful protein underpins the normal development of mammary glands and profoundly influences the cellular processes that sustain breast tissue (Table 2). The dual role of IGF-1 in maintaining normal breast physiology and driving BC pathology underscores its critical importance in both health and disease [15,38]. IGF-1 is crucial for the proper development of the mammary gland. In mice deficient in IGF-1 (Igf1−/−), there is a noticeable reduction in terminal end buds (TEBs), structures rich in mammary gland stem cells. This deficiency can be rescued by administering exogenous IGF-1, demonstrating its critical role in mammary gland development [13]. Studies using IGF-1 transgenic mice have shown that elevated levels of circulating and local IGF-1 enhance mammary gland development, further underscoring its importance [39]. Moreover, IGF-1 plays a crucial role in facilitating alveolar differentiation in the adult mammary gland during pregnancy and lactation [40]. Its interaction with ER-α underscores its significance in normal mammary gland development and function. ER-α, activated by estrogen binding, regulates gene transcription by binding to estrogen response elements (EREs) within gene promoters [41]. The influence of IGF-1 on ER signaling highlights its pivotal role in maintaining mammary gland biology and function [42].

In the context of BC, IGF-1 and IGF-1R have been extensively studied. IGF-1R localizes at the cell membrane, cytoplasm, and nucleus in normal breast tissue. Its subcellular localization has prognostic significance for BC [43]. Cytoplasmic IGF-1R staining in benign breast tissue is associated with a higher risk of developing BC. In ER+ tumors, cytoplasmic IGF-1R is linked with a longer disease-free survival (DFS), whereas in the TNBC subtype it has an unfavorable prognostic impact [10]. Recent observations suggest that nuclear IGF-1R plays a significant role in BC (Table 3). Nuclear IGF-1R staining is associated with more aggressive tumors and poorer survival outcomes [44]. IGF-1R’s nuclear translocation is closely associated with cell proliferation in multiple BC cell lines. Inhibiting IGF-1R nuclear localization reduces cell proliferation and migration in both non-malignant mammary cells and BC cells [45]. This suggests that nuclear IGF-1R controls fundamental cellular processes that are disrupted during malignant transformation. When in the nucleus, IGF-1R can directly bind DNA and regulate transcription [13]. IGF-1R binds to and stimulates its own promoter in the absence of ER expression in BC cells, indicating a unique role for IGF-1R in ER- BCs [46].

### 3.2. IGF-1 Isoforms and Receptor Interactions in Breast Cancer

IGF-1 and its isoforms play critical roles in BC development and progression. Each isoform has distinct functions, influencing various aspects of tumor biology [48]. The IGF-1Ea isoform has been shown to promote cell proliferation and survival. Elevated levels of IGF-1Ea correlate with the increased proliferation of BC cells [49]. Studies have demonstrated that the overexpression of IGF-1Ea in BC cells leads to enhanced tumor growth and survival, indicating its significant role in cancer progression [15,50]. The IGF-1Eb isoform, although less extensively studied, also contributes to BC. Research suggests that IGF-1Eb may promote tumorigenesis by enhancing cell proliferation and inhibiting apoptosis. Experimental models have shown that the overexpression of IGF-1Eb results in increased tumor growth and resistance to cell death, highlighting its potential as a therapeutic target [13,51,52]. As mentioned in the previous chapter, another crucial isoform is IGF-1Ec, IGF-1Ec is upregulated in response to mechanical stress and tissue damage [53]. In BC, the peptide PEc that defines the IGF-1Ec has been found to promote cellular proliferation migration and invasion by enhancing the EMT process, contributing to metastasis [54].

The significance of circulating IGF-1 levels in BC risk has been extensively studied [13,55,56]. Elevated serum IGF-1 concentrations have been linked to an increased risk of developing BC [57]. However, this relationship remains controversial, with some studies showing a stronger correlation in postmenopausal women than in premenopausal women [17]. Additionally, IGFBP-3, which modulates IGF-1 activity, has been associated with BC risk. High levels of IGFBP-3 have been linked to increased BC risk, although this association is not consistently observed across all studies [58]. IGF-1′s role extends beyond normal breast development to BC progression. The IGF-1 signaling pathway’s interaction with ER and other growth factors highlights its complexity and importance [2,59].

Furthermore, IGF-1 bioavailability in the bloodstream is modulated by IGFBPs, with IGFBP-3 being the predominant binding protein [10,60]. While IGFBP-3 can suppress BC cell growth and promote apoptosis in vitro, Mendelian randomization analyses indicate no significant link between genetically predicted IGFBP-3 levels and BC risk [61]. In contrast, several large-scale studies have consistently demonstrated a positive association between circulating IGF-1 levels and BC risk [62]. In BC cells, E2 treatment induced IGFBP-4, which subsequently inhibited E2-stimulated phosphorylation of ERα, the Wnt mediator GSK3β, and Akt. There was no evidence suggesting IGF1R’s involvement in the activation of Akt by E2 or its inhibition by IGFBP-4 [63]. A study indicated that IGFBP-4 could suppress E2-stimulated growth of BC cells through an IGF-dependent mechanism [52]. However, it is important to note that IGFBP-5, which might have a different IGF-independent mechanism from IGFBP-4, exhibited similar effects in this system [11,64].

The UK Biobank study, which included 4360 incident BC cases, found that higher IGF-1 levels were significantly associated with an increased risk of BC. This association remained robust across various subgroups, including different menopausal statuses and other risk factors [65]. Adjustments for other biomarkers, such as testosterone, sex hormone-binding globulin (SHBG), C-reactive protein (CRP), and hemoglobin A1c (HbA1c), did not alter the findings [66]. Notably, correcting for regression dilution bias, which accounts for measurement error and within-person variability, strengthened the observed association. This suggests that earlier studies might have underestimated the link between IGF-1 levels and BC risk [6]. For instance, the Endogenous Hormones and BC Collaborative Group reported an odds ratio (OR) of 1.25 for an 80th percentile difference in IGF-1 levels, whereas the hazard ratio (HR) in the UK Biobank analysis increased from 1.26 to 1.37 after corrections, underscoring the significant risk posed by elevated IGF-1 [67]. Moreover, Mendelian randomization (MR) analyses further supported these findings, revealing a positive association between genetically predicted IGF-1 concentrations and BC risk, specifically limited to ER+ tumors [68]. These results reinforce the causal role of the IGF pathway in the development of ER+ BC [65]. To provide a clearer overview of the current research on IGF-1 and its association with BC, Table 4 summarizes several key studies and findings related to circulating IGF-1 levels and their impact on BC risk and progression and Table 5 provides a detailed overview of the role of IGF-1 in breast cancer progression, summarizing both in vitro and in vivo evidence.

The IRS (insulin receptor substrate) proteins, which are essential in regulating the response of tumors to IGF signaling in BC, also impact patient outcomes [69]. IRS-1 is an ER-regulated gene with its expression highest in well-differentiated, ER+ luminal tumors and lowest in poorly differentiated, higher-grade tumors lacking ER expression. Conversely, IRS-2 is expressed at higher levels in ER-, basal/TN cells, and tumors [70]. The localization of IRS proteins also affects function and prognosis. IRS-1, found in the cytoplasm or nucleus, interacts with ER to regulate gene expression [69]. Nuclear IRS-1 is indicative of active ER signaling and is associated with a better clinical prognosis [71]. Membrane staining of IRS-2 is associated with a decreased overall survival (OR) in BC patients, particularly those with PR- tumors. The recruitment of IRS-2 to the plasma membrane after IGF-1R activation may explain this observation [72]. IRS-2 is also recruited to and activated by additional surface receptors, suggesting that membrane IRS-2 staining in BC could represent the activity of IGF-1R-dependent and -independent pathways [73].

The IR gene, consisting of 22 exons, undergoes selective splicing to form two IR isoforms: IR-A and IR-B [74]. These isoforms have different distribution patterns and exhibit significant differences in ligand-binding affinity for insulin and insulin-like growth factors, as well as in signal transduction pathways and biological effects [75]. Studies have shown that IR isoforms, particularly IR-A, may promote BC cell growth [48,74]. A higher IR-A/IR-B ratio is observed in BC tissues compared to normal breast tissues. This higher ratio is associated with a shorter DFS and poorer outcomes in human TNBC [55]. ER+ BC displays significantly higher levels of IR-A compared to ER− BC. In hormone therapy-resistant ER+ BC, there is an increase in both IR expression levels and the IR-A to IR-B ratio, along with a decrease in IGF-1R expression [56].

While elevated IGF-1 levels are linked to increased BC risk, the relationship between circulating IGF-1 and BC prognosis remains unclear, with inconsistent results [2,13]. A multiethnic, prospective cohort study involving 600 women with Stage I–IIIA BC found that high serum levels of IGF-1 and the IGF-1/IGFBP-3 ratio were associated with a higher risk of all-cause mortality. However, most other studies did not find a correlation between elevated levels of IGF-1 and IGFBP-3 with adverse BC outcomes such as all-cause mortality, BC-specific mortality, and recurrence [76,77]. In contrast, another large prospective cohort study reported that circulating IGF-1 was inversely and independently associated with all-cause mortality in invasive BC patients, consistent across various clinical risk factors [17,62].

**Table 4 ijms-25-09302-t004:** Summary of current IGF-1 research in breast cancer. This table provides an overview of recent studies on insulin-like growth factor 1 (IGF-1) in the context of breast cancer (BC). Note that this summary includes a selection of studies rather than a comprehensive review of all of the available research.

Study (Reference)	Research Focus	Key Findings	Clinical/Scientific Impact
Mauro De Santi et al., 2023 [78]	Association between IGF-1 levels, MetS, and insulin resistance in BC survivors	- Lower circulating IGF-1 levels in BC survivors with MetS compared to those without MetS - Interaction between HDL-C, glycemia, and IGF-1 levels, especially in subjects without MetS	- Highlights the impact of metabolic syndrome and insulin resistance on IGF-1 levels in BC survivors - Potential lifestyle interventions to modulate IGF-1 concentrations
Heba Mohammed Arafat et al., 2023 [76]	Investigate the relationship between IGF-1 and IGFBP-3 levels and BC risk among women in Gaza	- Age, lower physical activity, increased FBG, and IGF-1 levels were associated with an increased BC risk	- Identifies key risk factors for BC specific to the Gaza Strip population
Heidari S. et al., 2022 [79]	IGF-1 expression in serum and peritoneal fluid of BC patients	- Increased IGF-1 expression in the serum and peritoneal fluid of BC patients compared to controls	- Supports the role of IGF-1 in BC progression and metastasis
Lee et al., 2022 [13]	Prognostic value of IGF-1R across BC subtypes	IGF-1R expression was associated with poor survival across all BC subtypes and more pronounced in certain subtypes	- Highlights the functional significance of IGF-1R phosphorylation in BC prognosis
Biello et al., 2021 [16]	IGF Expression in relation to BC characteristics	- Higher IGF expression was associated with worse prognostic features in BC	- Provides insights into the association between IGF expression levels and the clinical characteristics of BC
Kamdje et al., 2021 [80]	IGF-1 and mammary tumor microenvironment	- Role of IGF-1 in regulating mammary tumor microenvironment; implications for tumor growth and metastasis	- Explores the influence of IGF-1 on tumor microenvironment dynamics and therapeutic strategies targeting IGF-1
Jennifer Tsui et al., 2021 [81]	IGF-1 and chemotherapy response in TNBC	- Study evaluating the role of IGF-1 in chemotherapy response and prognosis in TNBC	- Explores therapeutic implications of IGF-1 pathway inhibition in TNBC
Murphy et al. 2020 [65]	Mendelian randomization of circulating IGF-1 and BC risk	- Higher circulating IGF-1 levels associated with increased BC risk; a causal relationship was suggested	- Supports IGF-1 as a potential target for BC prevention strategies
Y Zhu et al., 2020 [77]	Association of circulating IGF-1 with BC mortality	- Inverse association between high circulating IGF-1 and all-cause mortality in BC patients	- Suggests a potential protective role of circulating IGF-1 in BC outcomes, especially in specific patient subgroups
Yiwei Tong et al., 2020 [82]	Evaluate the prognostic value of IGF-1 and metabolic abnormalities in HER2+ BC patients	- High IGF-1 levels are more common in pre/perimenopausal women and those with high IGFBP-3 - IGF-1 level was not associated with recurrence-free survival (RFS) in the whole population	- Identifies a subset of patients with distinct prognostic profiles based on IGF-1 levels - IGF-1 levels alone are not sufficient as a prognostic marker for RFS
Yujing Zhou et al., 2020 [83]	IGF-1 and estrogen (ERβ) in BC progression	- IGF-1 stimulates ERβ and aromatase overexpression in BC cells, promoting disease progression	- Highlights IGF-1′s role in promoting estrogen-driven pathways in BC
Gui-Ping Xu et al., 2018 [84]	Investigate the association between the rs1520220 polymorphism in the IGF1 gene and cancer susceptibility	- No overall positive association between rs1520220 and cancer risk	- Provides evidence that rs1520220 may not be a universal marker for cancer risk
H Li et al., 2016 [85]	Genetic polymorphisms in IGF-1 and BC risk	- Association between IGF-1 gene polymorphisms and BC risk; ethnic and subtype-specific differences	- Highlights genetic variations in IGF-1 and their impact on BC susceptibility across different populations
O’Flanagan et al., 2016 [86]	IGF-1R signaling and DNA damage response	- IGF-1R signaling sensitizes BC cells to cisplatin-induced DNA damage via ATM/ATR pathways, implicating IGF-1R in DNA repair mechanisms	- Highlights IGF-1R as a potential target for enhancing the DNA damage response in BC therapy
Christopoulos et al., 2015 [15]	Prognostic impact of IGF-1R expression among BC subtypes	- IGF-1R expression varies across BC subtypes; associated with a better BCSS in Luminal B but a worse outcome in the HER2-enriched subtype	- Highlights subtype-specific prognostic implications of IGF-1R expression, with potential for targeted therapies
Yerushalmi et al., 2012 [87]	Differential expression of tissue IGF-1 and BC subtypes	- Tissue IGF-1 levels were associated with a better prognosis in ER+ BC, in a conflicting role compared to serum IGF-1 levels	- Emphasizes the discrepancy between circulating and tissue-specific IGF-1 levels in BC prognosis
Litzenburger et al., 2011 [88]	IGF-1R inhibitors in BC cell lines	- Effective inhibition of IGF-1R with BMS-754807 in ΤΝBC cell lines, reversing IGF-1R-activated gene expression signatures	- Demonstrates the potential efficacy of IGF-1R inhibitors in ER-negative and HER2− BCs
Aleksic et al., 2010 [45]	IGF-1R nuclear translocation in BC	- Nuclear localization of IGF-1R was associated with aggressive tumor behavior and poor prognosis in BC	- Investigates the role of IGF-1R nuclear translocation as a prognostic marker and potential therapeutic target
Runhua Shi et al., 2004 [89]	Evaluate the association of IGF-1 and IGFBP-3 levels with BC risk through a meta-analysis	- Circulating levels of IGF-1 were significantly higher in premenopausal BC patients	- Supports the association between high IGF-1 levels and increased BC risk in premenopausal women.

**Table 5 ijms-25-09302-t005:** Summary of in vitro and in vivo evidence for IGF-1 in breast cancer proliferation, migration, invasion, and metastasis. This table consolidates data from studies utilizing breast cancer (BC) cell lines and xenograft animal models to explore IGF-1′s impact on key aspects of cancer behavior, including proliferation, migration, invasion, and metastasis.

Type of Evidence	Cell Lines	IGF-1 or IGF-1 Isoform of Interest	Method	Results	References
In vitro	MCF-7	Ec peptide	MTT assay/Trypan Blue	hEc in low doses stimulated the proliferation of wt cells	[59]
		rhIGF-1		Exogenous administration of human mature IGF-1 (rhIGF-1) in monolayer cultures stimulated the growth/metabolic activity of wt cells	
		Ec peptide	Wound healing/Scratch assay	MCF-7Ec cells have increased motility/migration capability compared to wt cells	
		Ec peptide	qPCR	An upregulation of Cdh-11, an indicative marker of EMT in MCF-7Ec cells, was detected	
	MDA-MB-231	Ec peptide	MTT assay/Trypan Blue	hEc did not stimulate the proliferation of wt cells	[90]
		rhIGF-1		Exogenous administration of rhIGF-1 in monolayer cultures stimulated the growth/metabolic activity of wt cells	
		rhIGF-1	Wound healing/Scratch assay	IGF-1 increased MDA-MB-231 cell migration approximately tenfold with half maximal stimulation at a concentration of 12 ng/ml	
In vivo	MCF-7	IGF-1	Mouse xenograft experiments	MCF-7 cells stably overexpressing IGF-1 induce significantly higher tumor volumes compared with control or mock cells	[15]
				Increased activation of the PI3K/AKT/mTOR pathway facilitated BCSC maintenance and increased their EMT phenotype	
	Immortalized human mammary epithelial cells	IGF-1R	Mouse xenografts experiments	Overexpression of IGF-1R caused cells to undergo EMT which was associated with dramatically increased migration and invasion	[91]
	MDA-MB-231	IGF-1/IGF-1R	TNBC patients	IGF-1 and IGF-1R overexpression were associated with an increased incidence of metastases and decreased survival	[92]

### 3.3. The Role of IGF-1 Signaling in Breast Cancer

The IGF signaling pathway, crucial for normal mammary stem cell regulation, also significantly impacts BC stem cells (CSCs) [13]. These CSCs, identified by surface markers such as CD133+, CD44+/CD24−, and CD29+, and functional markers like ALDH activity, play a key role in tumor initiation, progression, and recurrence [93]. CD44+CD24− CSCs display elevated IGF-1R expression, leading to increased stem cell activity, including mammosphere formation (clusters of mammary cells) [94]. In human BC xenografts, heightened IGF-1R phosphorylation signifies increased pathway activity. Targeting IGF-1R effectively reduces ALDH+ and CD44+/CD24− CSC populations, highlighting its importance in sustaining the CSC niche [95]. Conversely, the IR is not prevalent in CSCs, and IGF-1 and IGF-2 more strongly regulate CSC functions than insulin, underscoring IGF-1R’s pivotal role in maintaining BC stemness [13].

The IGF signaling pathway’s involvement in resistance to cancer therapy has also been investigated. The pathway modulates the expression of epidermal growth factor receptors (EGFRs) in cancer cells and may play a role in inducing resistance to endocrine therapies in ER+ BC patients [96]. Interactions between the IGF-1R and the HER2 pathway may contribute to resistance to anti-HER2-targeted therapy. IGF-1′s interaction with BC genes has been explored, particularly its relationship with the BRCA1 gene [16]. BRCA1, a transcription factor involved in DNA damage repair, cell growth, and apoptosis, is known for its role in hereditary BC. BRCA1 mutations are associated with a high cumulative risk of developing BC [82]. BRCA1 represses IGF-1R gene activity in BC cell lines, and BRCA1 deficiency leads to increased expression of the IGF system members [47]. IGF-1 increases BRCA1 gene expression and enhances BRCA1 promoter activity, indicating a complex, bidirectional interplay between IGF-1 pathways and BRCA1-mediated tumor protective pathways [97].

Further research into the molecular mechanisms underlying IGF-1′s role in BC has revealed distinct patterns in the expression of IGF-1 isoforms [43]. In vitro studies have shown that the IGF-1Ea and IGF-1Ec isoforms are expressed in both ER+ and ER− BC cells, while IGF-1Eb transcripts are absent. The recombinant rainbow trout (rt) pro-IGF1-E-peptides, analogous to human IGF-1 isoforms, exhibited anti-cancer activity in various human BC cell lines [22]. For example, the rtEa4-peptide inhibited MDA-MB-231 cell invasion by suppressing the activity of genes involved in classical signaling pathways such as PI3K, PKC, MEK1/2, JNK1/2, and p38 mitogen-activated protein kinase (MAPK) [47]. Similarly, the hEb peptide demonstrated anti-cancer properties by inhibiting tumor growth and angiogenesis in MDA-MB-231 cells [98]. In contrast, the hEc peptide promoted the proliferation and migration of ER+ BC cells via the ERK1/2 signaling pathway, suggesting a potential oncogenic role in hormone-sensitive BC [22]. Additionally, the IGF-1 pro-forms (IGF-1Ea, IGF-1Eb, and IGF-1Ec) were found to enhance BC cell proliferation and induce IGF-1R phosphorylation. Notably, anti-IGF-1R antibodies completely inhibited their activity, whereas anti-IGF1 antibodies only partially inhibited their biological effects [99].

These findings highlight the complex role of IGF-1 and its isoforms in BC, suggesting distinct biological functions and regulatory mechanisms [29,36]. While elevated IGF-1 levels are consistently linked to an increased BC risk, their impact on the prognosis and the treatment response requires further investigation [96]. The mechanisms by which IGF-1 exerts its effects on BC involve intricate processes of receptor binding, activation, and subsequent signal transduction pathways [82]. IGF-1 primarily functions through binding to the IGF-1R, a tyrosine kinase receptor located on the cell surface. Upon IGF-1 binding, IGF-1R undergoes autophosphorylation on specific tyrosine residues, initiating a cascade of downstream signaling events [8]. This receptor activation triggers two major pathways: the PI3K/AKT pathway and the MAPK/ERK pathway. The PI3K/AKT pathway plays a crucial role in cell survival and growth [100,101]. The activation of IGF-1R recruits and activates phosphoinositide 3-kinase (PI3K), which then generates phosphatidylinositol-3,4,5-trisphosphate (PIP3). PIP3 serves as a docking site for AKT (also known as protein kinase B), facilitating its phosphorylation and activation [101]. Activated AKT promotes cell survival by inhibiting apoptotic processes through the phosphorylation of downstream targets such as BAD, a pro-apoptotic protein, and by activating mTOR (mammalian target of rapamycin), which stimulates protein synthesis and cell growth [102]. In BC, dysregulation of the PI3K/AKT pathway is common, often due to mutations or amplifications in genes encoding components of this pathway, leading to enhanced cell survival, proliferation, and resistance to apoptosis [103].

The MAPK/ERK pathway is another critical signaling cascade activated by IGF-1R. Following receptor activation, the adaptor proteins IRS and SHC (Src Homology 2 domain-containing) are phosphorylated, leading to the recruitment and activation of the RAS protein [104]. Activated RAS then triggers a kinase cascade involving RAF, MEK, and ERK. ERK, upon activation, translocates to the nucleus and phosphorylates various transcription factors, including ELK1 and AP-1, which regulate the expression of genes involved in cell proliferation and differentiation [105]. This pathway is often upregulated in BC, contributing to increased tumor growth and progression [106].

In addition, IGF-1 signaling interacts with other growth factor pathways, including the epidermal growth factor receptor (EGFR) and HER2/neu (ErbB2) pathways. Crosstalk between IGF-1R and these receptors can enhance signaling output and contribute to therapeutic resistance [107]. For instance, IGF-1R activation can lead to the phosphorylation and activation of HER2, even in the absence of HER2 ligands, promoting cell proliferation and survival. This crosstalk is particularly relevant in HER2+ BCs, where co-targeting the IGF-1R and HER2 pathways may offer a more effective therapeutic strategy [17,108].

The regulation of IGF-1 and IGF-1R expression is complex and involves various transcriptional and post-transcriptional mechanisms. Estrogen, through its receptor ER, upregulates the expression of IGF-1R, highlighting the interplay between hormonal and growth factor signaling in BC [109]. Additionally, miRNAs play a significant role in modulating IGF-1 signaling. miRNAs are small non-coding RNAs that can bind to the 3′ untranslated region (UTR) of target mRNAs, leading to their degradation or translational repression [110]. Several miRNAs, such as miR-145 and miR-497, have been shown to target IGF-1R and downregulate its expression, thereby inhibiting IGF-1 signaling and reducing BC cell proliferation and invasion [110,111]. While some clinical trials have shown promising results, others have faced challenges, such as limited efficacy and resistance mechanisms [14,44,98]. Understanding the context-dependent roles of IGF-1 and its receptor, as well as their interactions with other signaling pathways, is essential for developing more effective and personalized therapeutic approaches.

## 4. The Role of IGF-1 in Breast Cancer: Mechanisms of Proliferation, Angiogenesis, Metastasis, and Resistance

### 4.1. Cell Proliferation and Survival

The role of IGF-1 as a critical growth and survival factor in human cancers is well-established. The IGF-1 gene generates various heterogeneous transcripts, all of which ultimately produce the mature form of IGF-1 [112]. In the tumor microenvironment, IGF-1 and its isoforms play significant roles in cell proliferation and survival. IGF-1 binds to the IGF-1 receptor, a tyrosine kinase receptor that transduces signals to the nucleus and mitochondrion primarily via the MAPK and PI3K/Akt pathways (Figure 1) [113]. IGF-1R activation leads to the recruitment and activation of PI3K which converts PIP2 to PIP3. PIP3 recruits Akt also known as Protein Kinase B to the cell membrane, where it is phosphorylated and activated by PDK1 and mTORC2 [17]. Activated Akt promotes cell survival by inhibiting pro-apoptotic factors, including Bad, and activating anti-apoptotic factors, such as Bcl-2 [114]. Akt also enhances cell proliferation by phosphorylating and inhibiting GSK-3β, leading to increased levels of cyclin D1 and cell cycle progression [115,116].

Recent studies have explored the effects of various peptides and proteins related to IGF-1. A synthetic PEc peptide, which comprises the final 24 amino acids of the translation product of the E domain of IGF-1Ec, is capable of stimulating the proliferation of breast, prostate, and osteosarcoma cancer cells and inducing the EMT process [103,106].

### 4.2. Angiogenesis and Metastasis

Angiogenesis is required for invasive tumor growth and metastasis and constitutes an important point in the control of cancer progression [116,117]. For tumors to develop in size and metastatic potential, they must make an “angiogenetic switch” through perturbing the local balance of pro-angiogenic and anti-angiogenic factors [118]. IGF-1 promotes the formation of new blood vessels, supplying the tumor with the essential nutrients and oxygen needed for growth and metastasis, by upregulating angiogenic factors like vascular endothelial growth factor (VEGF), through the activation of MAPK and PI3K/ AKT cascades [119]. Transgenic mice engineered to overexpress IGF-1 in their mammary epithelium show increased levels of VEGF in prepubertal glands and induce the expression of cyclooxygenase-2 (COX-2). COX-2 is an inflammatory molecule linked to angiogenesis and the production of prostaglandins (PG) [120]. Most BCs spread via the lymphatic system, and the presence of tumor cells in regional lymph nodes is a significant prognostic indicator for patients [121]. IGF-1 can stimulate the production of VEGF-C, a crucial pro-lymphangiogenic factor that promotes the proliferation and migration of lymphatic endothelial cells, thereby aiding BC metastasis [122].

Additionally, several BCs are invasive, with cells from primary tumors infiltrating local breast tissue, vasculature, and lymphatics, thereby advancing to metastatic disease [123]. The migratory and invasive capabilities of tumor cells result from a combination of intrinsic structural changes and extrinsic modifications to the surrounding extracellular matrix (ECM) [1]. Furthermore, IGF-1 induces MUC1 expression, a glycoprotein engaged in multiple cancer-related pathways, via AKT signaling, promoting the translocation of β-catenin and EMT progression in MCF7 cells [124]. Therefore, IGF-1 promotes the migration of BCs by interacting with the Wnt/β-catenin pathway, possibly involving other molecules, through AKT and/or ERK signaling [15]. IGF-1 administration triggers the invasion of MDA-MB-231 BC cells by stimulating the formation of lamellipodia, which are cellular protrusions at the leading edge of moving cells. These structures are supported to function as motors, propelling the cells forward during migration [125].

Moreover, the significant induction of migration in MCF-7Ec cells, along with the observed upregulation of pERK1/2 levels, aligns with studies on bone marrow-derived mesenchymal stem cells and tenocytes, which show actin filament formation and/or increased matrix metalloproteinase-2 (MMP-2) activity in response to hEc, likely via ERK1/2 signaling [126]. Cdh-11 expression has been previously associated with an aggressive BC cell phenotype and ability for bone metastasis. An upregulation of the Cdh-11 expression was noticed in MCF-7 hEc-overexpressing cells, which indicates hEc may be part of a metastatic process in BC cells [127].

BC metastasizes primarily to the lung, liver, brain, and bones, where IGF signaling is crucial for tumor growth and survival in these secondary sites [4]. IGF-1 and IGF-2 are known to enhance BC cell proliferation in vitro, and mouse models show that IGF-1R signaling promotes tumor growth in distant tissues [52]. Tumors with a dominant negative IGF-1R (dnIGF-1R) in the bone show reduced mitosis and increased apoptosis compared to those with functional IGF-1R [13]. Similarly, inhibiting IGF-1R with picropodophyllin (PPP) decreases BC proliferation in the brain. Mice treated with a combination of IGF neutralizing antibodies and chemotherapy have smaller lung metastases with fewer proliferating cells than those treated with chemotherapy alone [128].

IGF signaling not only enhances proliferation, but also influences other processes, including cancer stemness, epithelial–mesenchymal transition (EMT), and invasion, which are essential for metastatic tumor establishment [129]. Regarding EMT, both in vitro and in vivo studies have demonstrated that IGF-1 induces EMT, where BC cells gain mesenchymal traits, such as increased motility and a loss of cell adhesion [130]. These mesenchymal characteristics are associated with more aggressive tumors. BC cells exposed to IGF-1 exhibit reduced epithelial marker E-cadherin and increased mesenchymal marker vimentin, adopting a fibroblast-like morphology [131]. Recent studies indicate that IGF signaling can promote EMT through IGF-1R-dependent and -independent mechanisms, involving interactions with the IR-A isoform and IGF-2, which increases β-catenin levels, a key EMT mediator [132].

Communication between metastatic tumors and local tissues is vital for growth and survival within the metastatic niche [133]. In vivo models reveal that stromal cells, not tumor cells, are major sources of IGFs, with non-immune stromal cells and tumor-associated macrophages (TAMs) expressing Igf1 and Igf2 mRNA [134]. Cancer-associated fibroblasts (CAFs) also express Igf1 RNA in invasive lobular carcinoma models [1]. IGF ligands are produced by brain pericytes and resorbed bone in ex vivo studies, while BC cells express IGF-1R. This indicates that paracrine signaling between stromal-produced IGF ligands and IGF-1R on tumor cells is crucial for secondary tumor expansion [13].

### 4.3. Resistance and Apoptosis

The tumor microenvironment drives tumorigenesis and fosters resistance to anti-cancer drugs and protection against apoptosis [80]. Drug resistance significantly contributes to BC progression and relapse and is commonly observed in patients with advanced disease. Studies have established that the upregulation and overactivation of IGF-1R confers resistance to many chemotherapeutic regimens, including hormone therapy, targeted agents, and cytotoxic antineoplastic agents [13]. BC cells engineered to overexpress IGF-1R become resistant to the ER-targeting drugs tamoxifen and fulvestrant [135]. Additionally, xenograft tumors derived from tamoxifen-resistant BC cell lines show increased levels of phosphorylated IGF-1R [136].

The overexpression and activation of IGF-1R have also been demonstrated to counteract the anti-proliferative effects of trastuzumab (an HER2-targeted therapy), isoform-specific PI3K inhibitors, and radiation [137]. IGF-1R signaling diminishes chemotherapeutic responses in BC through various mechanisms, including promoting proliferation, enhancing cancer stemness, and inhibiting apoptosis via DNA damage repair [13]. The overexpression of hEc enhances the proliferation and migration of MCF-7 cells and increases the expression of ERα and the phosphorylation of ERK1/2 compared to wt MCF-7 cells. This, along with the preferential proliferation response to exogenous hEc in ER+ wtMCF-7 cells, suggests a potential correlation between hEc and estrogens in BC [90]. Although nearly 70% of BC cases express ERα, resistance to anti-estrogen therapy commonly develops [138].

Various mechanisms of hormone resistance involve ERK signaling and the reactivation or overactivation of ERα through enhanced IGF-1 signaling, indicating that hEc signaling might also contribute to the anti-estrogen-resistant phenotype of BCs [139]. Since IGF-1 signaling has previously been correlated with the chemoresistance of BCs, the response of the hEc overexpression model to a widely used chemotherapeutic agent showed that MCF-7Ec cells tended to resist docetaxel-induced apoptosis relatively more successfully, showing a slight increased survival in response to 50 nM docetaxel for 48 h, as compared to the docetaxel-induced apoptosis of mMCF-7 cells [140].

## 5. Clinical Implication and Therapeutic Potential

### 5.1. Therapeutic Targeting of IGF-1 and Its Isoforms

Both in vitro and in vivo evidence strongly suggest that IGF-1 system could serve as a promising target in BC therapeutics. Since IGF-1R activation relies on ligand binding and elevated IGF-1 levels promote tumor growth in BC, reducing IGF-1 levels is considered a viable anti-cancer strategy [141]. The secretion of GHRH by the hypothalamus triggers a cascade of molecular events, including the production of GH [142]. GH, in turn, stimulates the liver to secrete IGF-1 through its binding to GHRs. GHRH antagonists, for example JV-1-36 and JMR-132, have been shown to effectively inhibit the growth and metastasis of BC cells in vivo, reduce the viability of TNBC cells, and suppress tumor growth in mouse xenograft models [143].

As analyzed in previous chapters, IGFBPs normally regulate the IGF-1 bioavailability. By using recombinant IGFBPs tumor-promoting effects of IGF-1 signaling can be diminished. Both exogenous and endogenous IGFBP1 can suppress IGF-1-induced motility and growth of BCs in vitro and in animal models [144]. IGFBP-2 is often associated with oncogenic activities and its expression correlates with BC and mediates metastasis. However, a non-matrix binding and protease-resistant recombinant IGFBP-2 suppress tumor growth and angiogenesis in a mouse model through IGF-1 depended actions [145]. Recombinant human IGFBP-3 produced in rice was able to inhibit the growth of BC MCF-7 cells and colon cancer HT29 cells. Additionally, the endogenous expression of IGFBP3 influenced apoptotic responses induced by DNA-damaging agents in vitro, highlighting its role as a p53 target molecule [146]. Lastly, experimental cancer therapeutics are exploring synthetic molecules mimicking IGFBP’s IGF-1 binding properties, though their effectiveness is limited by IGF-1 independent and sometimes tumor-promoting actions of IGFBPs [147,148].

### 5.2. Monoclonal Antibodies and Small Molecule Inhibitors in IGF-1R Targeting

Another method to disrupt IGF-1R signaling is to eliminate free ligand levels by using neutralizing antibodies against IGF-1 [9]. Most are IgG1 or IgG3, which recruit immune cells to tumor sites, enhancing the anti-cancer response [149]. Human monoclonal antibodies (mAbs), such as MEDI-573, BI 836845, KM1468, and m708.5 block the binding of both IGF-1 and IGF-2 to IGF-1R and IR-A, resulting in suppressed proliferation and decreased phosphorylation of IGF-1R [150]. This technique offers an advantage over those directly targeting IGF-1R, as mAbs neutralize ligands outside the tumor microenvironment, while cell-related neutralization of the receptor allows for penetration within the tumor via receptor endocytosis and subsequent degradation in the lysosome [9]. With over forty clinical trials focusing on IGF-1R in BC, IGF-1R remains a significant target in BC therapeutics [95]. Among the main strategies for inhibiting IGF-1R, mAbs have been extensively tested. These antibodies inhibit IGF-1R activation by blocking ligand binding, specifically targeting the IGF-1R/InsR hybrid, but not the InsR homodimer. Notable antibodies include figitumumab, ganitumab, and dalotuzumab [46].

Figitumumab, an IgG2 mAb, was tested in BC clinical trials to block IGF-1R signaling. Although initial results were promising, particularly in combination therapies, figitumumab ultimately failed to improve PFS or OS in advanced stages. Common adverse events, such as hyperglycemia, limited its further development [128]. Dalotuzumab, another mAb, specifically targets the IGF-1R alpha subunit, preventing ligand binding and receptor activation. Despite early success in trials, it did not show significant therapeutic benefit in larger studies and faced challenges due to adverse effects and limited efficacy [51]. Additionally, Ganitumab, an IgG1 mAb, demonstrated notable clinical activity in early-phase BC trials. It has shown potential in combination therapies, such as with the chemotherapeutic agent gemcitabine, improving overall survival in metastatic BC patients [150]. Ganitumab’s acceptable toxicity profile and ongoing trials, including combinations with palbociclib and dasatinib for Ewing’s sarcoma, highlight it as a promising therapeutic agent for BC [151]. These antibodies offer greater specificity for IGF-1R with fewer metabolic side effects compared to other inhibitors. They enhance the anti-cancer response by recruiting immune cells to tumor sites, providing a multifaceted approach to targeting IGF-1R in BC (Table 6) [150].

### 5.3. Emerging Therapies and Novel Approaches in IGF-1 and Its Receptor Inhibition

An further crucial treatment approach uses small molecules targeting the receptor’s tyrosine kinase domain [162]. These may have improved overall anti-cancer efficiency as they can inhibit the tumor-promoting effects mediated by IGF-2 signaling and hybrid IGF-1R/IR receptors. Preclinical studies have demonstrated that several dual IGF-1R/IR tyrosine kinase inhibitors (TKIs), such as OSI-906 (linsitinib), BMS-554417, and BMS-754807, show anti-proliferative and pro-apoptotic effects in BC cells [163,164]. These inhibitors also effectively reduce BC growth in vivo, whether administered alone or in combination with hormone therapy. Phase I dose escalation trials of OSI-906 and BMS-754807 demonstrated that the use of TKIs resulted in hyperglycemia but was ultimately tolerable and potentially had anti-tumor activity [165]. BMS-536924, a dual IGF-1R/IR TKI, has shown promise in inhibiting the growth of various BC cell lines in preclinical studies, but clinical evidence is still lacking [166]. While both approaches have shown experimental promise, the clinical trial results have been disappointing, often due to severe side effects and limited efficacy (Table 5) [18].

Oligonucleotide therapy has gained attention as a potential novel approach for cancer treatment, primarily targeting oncogenes through gene depletion. Additionally, this strategy is being evaluated for its immunostimulatory effects [167]. Cancer immunotherapy, particularly personalized treatments like autologous cell vaccines, utilizes a patient’s own tumor cells to elicit a cytotoxic T-cell response against tumor-specific antigens [168]. Research involving BC and glioblastoma multiforme (GBM) models has demonstrated that antisense oligonucleotides (AS-ODN) targeting IGF-1R can provoke anti-tumor immune responses. The immunostimulatory effects are largely due to the unmethylated CpG motifs in AS-ODN, which activate Toll-like receptor 9 (TLR9) and trigger both innate and adaptive immune responses [150]. In BC, IGF-1R inhibition led to the release of proinflammatory cytokines (TNF-α and IFN-γ) and was associated with CD8+ T-cell-mediated cytotoxicity through the Fas/FasL pathway. In GBM, the response also involved B cells and tumor-specific antibody production [164].

Other approaches using nucleic-acid based strategies have been used to investigate the IGF-IR/IGF-I pathway, including ribozymes, triplex-forming oligonucleotides, and short interfering RNAs (siRNAs) [169]. The in vivo administration of phosphorothioate antisense oligonucleotides targeting IGF-IR decreased receptor protein levels and concomitantly inactivated AKT and MAPK signaling pathways leading to C4HD BC growth inhibition [169]. SiRNA-based therapies are being evaluated clinically; however, there are still major barriers to therapeutic application [150].

Apart from the above therapeutic strategies, antibody–drug conjugates (ADCs) are a significant focus in oncology therapeutics, with around 80 ADCs in clinical trials and 4 approved by the FDA and European Medicines Agency for metastatic cancers [170]. ADCs have been exploited to treat IGF-1R overexpressing tumors. A new ADC, W0101, combines a mAb (hz208F2-4) with a cytotoxic derivative of auristatin using a non-cleavable linker [171]. W0101 shows high specificity for IGF-1R and induces potent cytotoxic activity in the IGF-1R-overexpressing BC cell line MCF-7 by blocking cell division through tubulin polymerization inhibition. A Phase I/II clinical trial is assessing the safety and efficacy of W0101 in advanced solid tumors (NCT03316638) [17].

Peptide–drug conjugates also target IGF-1R-overexpressing tumors. An engineered IGF-1 variant (765IGF) with a reduced affinity for IGFBPs is conjugated with methotrexate (MTX) to form IGF-MTX. This conjugate selectively enters cancer cells via IGF-1R, showing promising anti-tumor activity in MCF-7 xenografts with fewer side effects compared to free MTX [172]. Two Phase I trials (NCT02045368 and NCT03175978) demonstrated promising efficacy and tolerable toxicity profiles. To enhance ADC efficacy, chemical modifications such as the PEGylation of the cytotoxic drug maytansine (DM1) have been explored [173]. This modification allows the anti-IGF-1R antibody cixutumumab conjugated to PEGylated DM1 to evade drug efflux pumps, increasing potency [174].

IGF-1R overexpression in BC can confer resistance to radiotherapy, chemotherapy, and targeted therapies. This resistance is often mediated by crosstalk with other signaling pathways and enhanced DNA repair mechanisms [51]. IGF-1R in the nucleus may maintain cancer stemness by activating WNT signaling, contributing to therapeutic resistance. Combining IGF-1R inhibitors with other treatments has shown promise [175]. For instance, co-treatment with IGF-1R inhibitors and EGFR inhibitors like gefitinib has reduced proliferation and invasion in resistant prostate cancer cells. In BC, high circulating IGF-2 levels correlate with a poor response to trastuzumab [51,150]. Combining IGF-1R inhibitors with trastuzumab, tamoxifen, or fulvestrant has shown enhanced anti-tumor effects in preclinical studies. Given that IGF-1R inhibitors can cause hyperglycemia, combining these with metformin, which has both hypoglycemic and anti-cancer effects, has shown superior therapeutic outcomes in BC cell lines [176]. Targeting the tumor microenvironment is another strategy. Co-targeting IGF-1R and STAT3 can inhibit tumor metastasis and overcome resistance to anti-IGF-1R therapy by preventing pro-angiogenic cytokine production and tumor angiogenesis [13]. Consequently, identifying biomarkers to predict the treatment response and understanding the tumor microenvironment and resistance mechanisms are crucial. Breakthroughs such as CRISPR-Cas9 are helping to develop more efficient cancer models for research [59,177].

## 6. Future Directions in Research

Future research in the IGF-1 signaling pathway and its implications for BC treatment holds significant promise. Despite strong evidence that IGF signaling promotes BC progression, clinical trials targeting IGF-1R and its ligands have often failed [150]. A major issue is the disruption of normal glucose uptake in peripheral tissues, raising plasma glucose levels and stimulating insulin and IGF-1 secretion, which compensates for the inhibited IGF-1R pathway and reduces therapies’ effectiveness [22]. To address this, future treatments must focus on maintaining normal glucose uptake and reducing peripheral glucose levels [178].

Recent studies have shown the potential of dietary interventions to lower glucose levels. Fasting and low-carbohydrate diets reduce basal glucose levels, leading to decreased insulin and IGF-1 release, enhancing the efficacy of glucose uptake-reducing therapies [179,180]. For example, preclinical studies demonstrated that lowering environmental glucose enhances the effects of traditional chemotherapy [181]. BC cells treated with doxorubicin, cyclophosphamide, and cisplatin showed increased DNA damage in low-glucose conditions. Mice with ER+ BC responded better to hormone therapy when fed a fasting-mimicking diet (FMD) compared to a regular diet [182].

These promising results have led to clinical trials investigating dietary approaches. A Phase II trial (NCT02126449) assessed the impact of an FMD on the response to neoadjuvant chemotherapy in HER2− BC, finding that patients on an FMD had a higher rate of complete or partial response to radiation [183,184]. A randomized Phase III trial has been registered to follow up on these findings. Additionally, another Phase II trial is recruiting patients to evaluate the anti-tumor effects of an FMD and metformin in TNBC, highlighting the growing interest in restrictive diets as a treatment strategy for various BC subtypes [13]. Maintaining a restrictive diet throughout treatment remains a challenge, but trial outcomes will provide valuable insights into whether dietary strategies can be a viable therapeutic option, either alone or in combination with existing IGF-1R targeting therapies [2,150].

In addition to dietary strategies, novel technological approaches are being explored to target the IGF/insulin systems. SiRNA and miRNA technologies show potential in reducing IGF-1R expression and function [10,185]. Durfort and colleagues demonstrated that silencing IGF-1R with synthetic siRNA containing 2′-O-methyl nucleotides can induce cell-cycle arrest and decrease cell proliferation [17,169]. Additionally, this study highlighted the potential of the IGF-I axis in mobilizing pro-inflammatory cytokines, offering a new clinical approach for treating mammary tumors expressing IGF-1R [17].

Despite the promise of siRNA formulations, systemic application faces challenges such as delivery to target the cell cytoplasm and the transient inhibition of the IGF pathway [185,186]. Preclinical in vivo studies suggest that these obstacles can be overcome through the stable and inducible long-term expression of target short hairpin RNA using drugs such as doxycycline or tetracycline [10]. Additionally, other miRNAs have been investigated. For example, decreased levels of miR-139, which targets IGF-1R in colorectal cancer, are associated with disease progression and metastasis [187]. Future modifications to siRNAs targeting IGF-1R may improve their effectiveness in downregulating IGF-1R and modulating anti-tumor immune responses, potentially offering new clinical approaches for treating mammary tumors expressing IGF-1R [149].

Inhibition of the GH/IGF-1 axis is also highlighted as a key strategy to enhance human health span by reducing the incidence of chronic age-related diseases, such as cancer and diabetes [188]. Research has made significant advances in understanding the endocrine functions of IGF–IGFBP complexes [16]. However, the details of IGFBP-dependent cell signaling have been slower to emerge [11]. Genetic and proteomic screens have identified many unexpected IGFBP ligands, revealing that their influence extends from the extracellular space to the cytoplasm and the cell nucleus. Despite recent discoveries of novel IGFBP ligands, knowledge of most protein–protein interactions involved in IGFBP signaling remains limited [189].

Targeting IRS proteins represents another unexplored approach for inhibiting the IIS pathway in tumors. IRS proteins are commonly overexpressed in BCs, and the in vitro knockdown of IRS expression limits BC cell proliferation and invasive potential [190]. While inhibiting both IRS adaptors would likely alter metabolic signaling pathways, antineoplastic therapies targeting either IRS-1 or IRS-2 might preserve normal host metabolism [13]. Targeting IRS-2 is particularly intriguing as it is highly expressed in more aggressive BCs for which there are few effective treatments [104]. A recent study identified a unique region within the IRS-2 C-terminal tail necessary and sufficient for BC cell invasion, but not required for IRS-2-dependent regulation of glucose uptake. This suggests that the role of IRS-2 in BC progression can be separated from its role in glucose metabolism [69].

The rationale behind developing therapeutic strategies in breast cancer, targeting signaling components of the IGF pathway, derives from the evidence supporting the direct and intricate implication of the IGF-1 and IGF-2 ligands and their receptors, mainly IGF-1R, in breast cancer initiation and progression. The major candidates for evaluation as anti-cancer agents are, anti-IGF-1R monoclonal antibodies, small molecule tyrosine kinase inhibitors and inhibitors of IGF ligands and their isoforms. However, although attractive in theory, the outcomes of clinical trials remain conflicting due to either poor efficacy or major adverse side effects due to the crosstalk between intertwined pathways. Selective targeting of IGF-1R, through preventing the receptor/ligand interaction or by decreasing the number or the receptor’s copies on the cell surface, disrupts the IGF-1, IR and growth hormone homeostasis. Growth hormone elevation, subsequent to the IGF-1R antibody, resulted in hyperglycemia and metabolic syndrome due to insulin resistance. Moreover, IR compensates for the loss of IGF-1R efficacy mainly due to their high homology in function and ligands. The same similarities in binding affinity between IGF-1R and IR are responsible for the failure of exclusive targeting of IGF-1R by small molecular inhibitors and the simultaneous limiting of IR signaling and compensation. This lack of selectivity might be promising but with the expense of metabolic side effects and increased toxicity. Neutralizing antibodies against IGF ligands do not influence receptor function and metabolic landscape, however, the preliminary results from large clinical trials are not encouraging at least with respect to breast cancer patients PFS.

Understanding the specific roles of the peptides that define the different IGF-1 isoforms in BC progression will be crucial. Each isoform may have distinct functions in various tissues and developmental stages, potentially influencing specific cellular pathways and disease processes. The fact that PEc levels of expression increase in in vivo models of various cancers indicates the vitalness of this molecule in BC development and progression. The PEc expression may also explain the reduced efficiency of the anti-IGF-1 and anti-IGF-1R mAbs in various cancers [191]. Recent evidence suggests that IGF-1Ec is produced when tumors are in hostile conditions (such as the parenthetic pressure of the surrounding environment applied to the tumor as the tumor grows, leading to the death of the tumor and the exposure of tumor antigens to the immune system) [192]. IGF-1Ec production after proteolytic cleavage leads to the release of the mature IGF-1 and PEc, both molecules inducing tumor progression via different receptors. According to this, the targeting of IGF-1 or IGF-1R will not stop the action of PEc, which apart from inducing cellular proliferation and EMT leading to metastasis and invasion also induces the recruitment of mesenchymal stem cells prior to tumor repair [112,192]. A mAb against PEc has been developed and has been applied in various cancers (prostate, breast, and colon) with spectacular results, underlining the importance of PEc targeting in cancer treatment [191].

These insights emphasize the potential for personalized medicine approaches in BC treatment. By selectively targeting tumor-specific functions of the IRS proteins without negatively impacting the metabolic functions of normal tissues, new therapeutic strategies can be developed. Personalized medicine could involve tailoring specific inhibitors to individual patients’ genetic and molecular profiles, thereby improving treatment efficacy and minimizing side effects.

## 7. Discussion

This review explores the functions of IGF-1 isoforms, IGF-1R, IGFBPs, and IR in BC, focusing on their roles in regulating cell growth, survival, and progression. The receptor system is intricate, as the IR and IGF-1R genes have the capacity to generate various hybrid receptors [8]. The metabolic cascade leading to hyperglycemia must be circumvented in order for the enormous potential clinical impact to be gained. Combining multiple agents and fine-tuning selective inhibitors which spare IR while impairing the pro-cancer functional significance of IGF-1 might be accomplished by further examining the structural differences and binding peculiarities of IGF-1R and IR. The functional overlap of the receptors involved as well as the composition of the hybrid receptors which participate in the signaling system emphasizes the importance of unwinding the role of downstream signaling molecules like signal adaptor proteins and developing inhibiting mechanisms which minimize unwelcome secondary receptor activation. The incorporation of pharmaceutical agents that normalize cell glucose uptake and plasma glucose levels in combination with IGF-1 axis blocking regimens might offer a degree of flexibility against adverse events. Patient stratification and a detailed examination of tumor biomarker and molecular profile could possibly identify responsive subpopulations. With at least 40 studies simultaneously recruiting patients, internal antagonism and weak study groups lead to inconclusive results. Perhaps an ally and incorporation of these studies into a large multicenter one would allow the patients included to be better cohosted and more useful results to be drawn.

This complexity underscores the need for a nuanced understanding of how these receptor configurations influence cellular responses to ligand activation [10]. Despite extensive research, clinical efforts to target the IGF signaling pathway have yielded disappointing results, highlighting the need for a deeper understanding of the molecular mechanisms by which IGF signaling drives BC [11,16,43]. Identifying improved biomarkers to accurately pinpoint responsive patient populations and discovering new targets that do not disrupt normal glucose homeostasis are essential steps toward harnessing this elusive target for clinical benefit [22].

A growing body of evidence supports the association of the IGF-1 system with BC establishment and progression [2,13,18]. Conflicting results arise from differing methodologies, distinct molecular subtypes studied, genetic variations between populations, and tumor heterogeneity [17]. Nevertheless, substantial experimental evidence implicates the IGF-1 system in BC biology. Efficiently targeting this system remains an attractive prospect, as further insights into its molecular mechanisms could pave the way for new diagnostic and treatment strategies [36].

In more detail, one major limitation of the existing research is the variability in study designs and methodologies, which can affect the reproducibility and generalizability of the results. For example, differences in experimental models, such as variations in the cell lines or animal models used, may influence the observed effects of IGF-1 signaling. Additionally, clinical studies often face challenges related to patient heterogeneity, including variations in genetic background, tumor subtype, and prior treatments, which can complicate the interpretation of findings and the efficacy of targeted therapies. Moreover, many studies have focused on isolated aspects of the IGF-1 signaling pathway, potentially overlooking complex interactions with other molecular pathways, such as ER signaling or immune response mechanisms. Finally, the impact of genetic and environmental variables on IGF-1 signaling and BC progression is not always adequately accounted for in studies. Addressing these limitations requires a multi-faceted approach, including the use of standardized methodologies, larger and more diverse patient cohorts, and integrative research strategies that consider the broader molecular context of BC [15].

The variety of IGF-1 gene transcripts and precursor forms used in forming mature IGF-1 presents a complex picture. While their exact roles in carcinogenesis remain unclear, available knowledge suggests that both transcripts and numerous IGF-1 precursor forms might exhibit carcinogenic activity, sometimes with opposing effects [43]. It is crucial to describe cancer-specific molecular mechanisms, especially those associated with the IGF-1Eb isoform and receptors present on the cell membranes of common human tumors [193]. In the nearest perspective, identifying candidate therapeutic targets for various IGF-1 isoforms, including Eb peptide analogues with anti-cancer activities, holds promise [22]. For example, an inhibitor of IGF-1Ec, known for its pro-proliferative potential in several cancers, could be highly therapeutic. However, classical anti-GH-IGF-1R therapies are inefficient for completely inhibiting the IGF-1Ec isoform, necessitating new approaches [191]. Recent studies have shown promise with PEc mAb, a mAb targeting the pro-proliferative IGF-1Ec isoform, indicating a potential new therapeutic avenue for BC treatment. This underscores the importance of further investigating PEc and PEc mAb as future therapeutic targets [191,192].

The clinical implications of IGF-1 research extend beyond its direct role in tumor biology, offering potential for broader therapeutic applications in BC management. One promising area is the use of IGF-1 pathway inhibitors in combination with other targeted therapies, such as PI3K/AKT/mTOR inhibitors, which are already being explored in clinical trials for their synergistic effects [150,188]. Additionally, IGF-1 signaling is implicated in the maintenance of cancer stem cells, which are often resistant to conventional therapies and contribute to tumor relapse. Targeting IGF-1R may therefore enhance the effectiveness of treatments aimed at eradicating these stem cells, potentially reducing recurrence rates [94,175]. Furthermore, IGF-1R’s involvement in modulating the tumor microenvironment, including interactions with immune cells, suggests that combining IGF-1 inhibitors with immunotherapies could bolster anti-tumor immune responses, offering a multi-pronged approach to combat BC. Finally, given the pathway’s role in metabolic regulation, exploring IGF-1 inhibitors in conjunction with lifestyle interventions, such as diet and exercise, could provide an integrated strategy to improve patient outcomes [149].

Given the complexity of the IGF-1R/IR family and the dynamic predominance of specific receptors and ligands in individual tumors, each class of anti-IGF/IGF-1R agents may offer unique advantages and different toxicity profiles in selected tumor settings [164]. Clinical data with anti-IGF-1R mAbs and TKIs show that these targeting approaches are feasible and can induce strong anti-tumor activities in several tumor types, including rare tumors refractory to standard therapies [51,194]. Clinical trials exploring dietary interventions and innovative technologies, such as siRNA and miRNA targeting IGF-1R and IRS proteins, offer promising strategies for enhancing BC treatment efficacy [16,164].

However, developing targeted therapies against the IGF-1 signaling pathway in BC presents several challenges that must be addressed to translate preclinical promise into effective clinical treatments. One significant challenge is the pathway’s complex biology, including the diverse roles of IGF-1 isoforms, hybrid IGF-1R/IR, and interactions with other signaling pathways. The IGF-1 signaling system is integral not only to tumor growth but also to maintaining normal cellular functions. Consequently, targeting IGF-1R or its ligands may lead to unintended side effects, such as hyperglycemia, complicating patient management and limiting the dosage of therapies. Another difficulty lies in the heterogeneity of BC itself, with different subtypes exhibiting varying levels of IGF-1R expression and pathway activity, suggesting that a one-size-fits-all approach will likely be ineffective. Personalized medicine strategies, including identifying biomarkers that predict which patients are most likely to benefit from IGF-1R-targeted therapies, are essential.

Furthermore, compensatory signaling mechanisms, such as the upregulation of alternative growth factor receptors or enhanced DNA repair pathways, may reduce the effectiveness of IGF-1R inhibitors. Drug resistance further complicates therapy development. Combining IGF-1R inhibitors with other therapeutic agents, such as EGFR inhibitors or hormone therapies, has shown promise in preclinical studies. However, the complexity of drug interactions and the potential for additive toxicities must be carefully managed [150]. Moreover, the clinical translation of promising preclinical findings is often hindered by difficulties in optimizing drug delivery and formulation. For example, small molecules and mAb targeting IGF-1R require effective delivery systems to ensure they reach and penetrate the tumor tissue effectively [195]. Additionally, the development of resistance to targeted therapies and the management of side effects like hyperglycemia underscore the need for ongoing research to refine treatment protocols and enhance the therapeutic index.

Despite promising results, these therapies are currently effective only in a subset of patients, underscoring the need for rational combination strategies and a deeper understanding of IGF-1R biology. Emerging research suggests that IGF-1R is crucial for stem cell maintenance and expansion and is expressed in more differentiated populations, implying that IGF-1R may drive stem cell characteristics, leading to increased therapy resistance, and influence lineage-linked traits, contributing to tumor heterogeneity [47,94]. Finally, the IR undergoes complex regulatory processes during selective splicing, resulting in distinct isoforms, IR-A and IR-B, which exhibit divergent functionalities in pathological conditions, such as cancer [32,74].

## 8. Conclusions

In conclusion, despite extensive efforts to target this pathway therapeutically in BC patients, significant challenges persist, emphasizing the critical need for further research to elucidate the precise molecular mechanisms through which this signaling pathway promotes BC progression. Advancing our understanding of the IGF-1 signaling pathway and its isoforms in BC is crucial for developing personalized medicine approaches. Tailoring therapies to individual patient profiles, optimizing targeting strategies, and elucidating the roles of different IGF-1 isoforms are pivotal steps toward improving clinical outcomes for BC patients. Future perspectives include the exploration of PEc and PEc mAb, which offer promising new avenues for therapeutic intervention by specifically targeting the IGF-1 isoforms associated with BC progression. This targeted approach may pave the way for more effective treatments and improved patient outcomes.

## Figures and Tables

**Figure 1 ijms-25-09302-f001:**
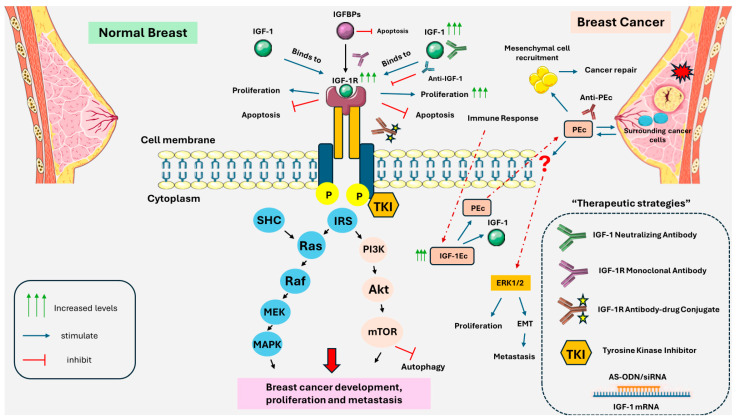
The Role of IGF-1 Signaling in Breast Cancer Development and Metastasis. This figure comprehensively depicts the role of IGF-1 signaling in breast cancer (BC) development and outlines some potential therapeutic strategies. On the left, the figure represents the morphology of a normal breast, while on the right, it illustrates a breast with cancerous changes. Central to the figure is the IGF-1 ligand and its receptor (IGF-1R), highlighting their interactions and the modulation by IGFBPs. IGFBPs strongly bind to IGF ligands, affecting their interaction with receptors. The figure further elaborates on the intracellular signaling cascades triggered by IGF-1 binding, including the PI3K-AKT pathway and the Ras-Raf-MEK-MAPK pathway, which are crucial for cell proliferation, survival, and metastasis. It also shows the transmembrane signaling components and cytoplasmic effectors involved in these pathways. Moreover, the figure delineates current therapeutic strategies targeting the IGF-1 signaling axis in BC. These include monoclonal antibodies (mAbs) designed to inhibit IGF-1R, kinase tyrosine inhibitors (TKIs) that block downstream signaling kinases, and antisense oligodeoxynucleotides (AS-ODNs) and small interfering RNAs (siRNAs) that specifically knock down key molecules in the signaling pathways. A key feature of the figure is the depiction of IGF-1 signaling within the cell. Here, IGF-1 is represented as IGF1-Ec, which upon degradation, releases IGF-1 and PEc. PEc interacts with an unidentified receptor, subsequently activating the ERK1/2 pathway involved in proliferation and epithelial–mesenchymal transition (EMT). This signaling also facilitates the recruitment of human mesenchymal stem cells (HMSCs) to the site, promoting tissue repair and suggesting a mechanism analogous to what has been observed in prostate cancer. Anti-PEC antibodies target PEc, preventing it from interacting with its receptor and thereby inhibiting the ERK1/2 pathway and subsequent HMSC recruitment, and anti-IGF-1 antibodies prevent IGF-1 from binding to IGF-1R and thus inhibit the activation of downstream signaling pathways.

**Table 1 ijms-25-09302-t001:** “IGF-1 Isoforms an Overview”: This table summarizes the structural features, generation mechanisms, functional differences, primary signaling pathways, alternative splicing patterns, and predominant tissue expressions for each isoform of IGF-1 (Ea, Eb, and Ec).

Isoform	Structure	Generation Mechanism	Alternative Splicing	Functional Characteristics and Differences	Primary Signaling Pathways	Predominant Tissues
IGF-1Ea	Exons 1, 2, 3, 4, 5	Splicing of exons 1–5 from the igf-1 gene	Modulates exon 5 inclusion/exclusion	- Similar to hepatic IGF-1, involved in systemic growth regulation - Promotes cell proliferation, survival, and metabolic functions - Interacts with IGFBPs to regulate bioavailability and activity	- PI3K/Akt, - MAPK (ERK1/2), - JAK/STAT - IGF-1R - IGFBPs - The specific roles and predominant tissues of each IGF-1 isoform can vary depending on developmental stage, physiological conditions, and species-specific differences.	- Liver, muscle, bone, brain, and adipose tissue
IGF-1Eb	Exons 1, 2, 3, 4, 6	Splicing of exons 1–4, 6 from the igf-1 gene	Excludes exon 5	- Predominantly expressed in liver, role in growth regulation - Specific role in growth modulation, potential interactions with IGFBPs		- Liver, muscle, and testes - Other Tissues: Present but with less clear predominant roles compared to IGF-1a and IGF-1c
IGF-1Ec	Exons 1, 2, 3, 5, 6	Splicing of exons 1, 2, 3, 5, 6 from the igf-1 gene	Excludes exon 4	- Response to mechanical stress in tissues - Enhances cell migration, tissue remodeling, and repair processes - Cleaved COO-terminal peptide (PEc) induces cellular proliferation and epithelial to mesenchymal transition (metastasis in prostate cancer)		- Muscle, heart, bone, and tendons

**Table 2 ijms-25-09302-t002:** “Insulin-like Growth Factor 1 (IGF-1) Expression in Relation to Breast Cancer (BC) Characteristics”: This table provides insights into the expression of Insulin-like Growth Factor 1 (IGF-1) across various patient groups with Breast cancer (BC). It highlights how IGF-1 expression varies based on estrogen receptor status, tumor stage, and lymph node involvement.

Patient Group	IGF-1 Expression (%)	Estrogen Receptor Status
ER+ (Positive)	60%	IGF-1R is often associated with ER+ BC
Triple-Negative (TNBC)	20%	IGF-1R expression tends to be lower in TNBC compared to ER+ subtypes
**Tumor Stage**		
Early (T1–T2)	35%	Higher IGF-1 expression correlates with more advanced BC stages (T3–T4)
Advanced (T3–T4)	65%	
**Lymph Node Status**		
Metastasis Present	60%	IGF-1 expression is higher in BC patients with lymph node metastasis
No Metastasis	40%	Lower IGF-1 expression is observed in BC patients without lymph node involvement

**Table 3 ijms-25-09302-t003:** “IGF-1R Expression Across Breast Cancer Subtypes”: This table highlights the varying levels of IGF-1R expression and their implications for prognosis and treatment response in Luminal A, Luminal B, HR+/ERBB2+, and Triple-Negative Breast Cancer (TNBC) subtypes [47].

BC Subtype	Percentage of IGF-1R Expression	Key Findings
Luminal A	~52%	IGF-1R expression does not affect breast cancer specific survival.
Luminal B	~57.5%	Higher total IGF-1R levels correlate with a better prognosis.
HR+/ERBB2+	~10–20%	Active phosphorylated IGF-1R/IR does not correlate with prognosis in trastuzumab-treated ERBB2+ tumors.
TNBC	~22–46%	IGF-1R expression correlates with shorter survival. TNBCs are responsive to IGF-1 signaling promoting proliferation and survival.

**Table 6 ijms-25-09302-t006:** Therapeutic approaches targeting the IGF-1 system in breast cancer. This table outlines various therapeutic strategies targeting components of the IGF-1 system in different subtypes of breast cancer (BC).

Target Component of the IGF-1 System	Details of Agent	BC Subtype	Phase of Development	Status	Combination Therapy	References
GH-RH	Antagonists JV-1-36 or JMR-132	TNBC	II	Completed	Docetaxel	[143]
	Somatostatin analog: Octreotide	Stage I, II, III BC HER2+ metastatic BC	ΙΙ	Completed Ongoing	Tamoxifen, ovariectomy	[152]
	Somatostatin analog: Pasireotide	Stage I, II, III BC	I	Ongoing	-	[153]
IGF-1	Neutralizing monoclinic antibody: MEDI-573	Hormone-sensitive, HER2− metastatic BC	II	Ongoing	Aromatase inhibitor	[154]
	mAb: BI 836845	Hormone-sensitive, HER2− metastatic BC	I	Ongoing	Everolimus, Exemestane	[155]
IGF-1R	mAb: CP-751.871	Postmenopausal subjects with advanced HR+ BC	II	Terminated	Exemestane, fulvestrant	[156]
	mAb: AMG-479	Postmenopausal subjects with locally advanced or metastatic HR+ BC	II	Completed	Exemestane, fulvestrant	[157]
	mAb: IMC-A12	HER2 positive advanced BC resistant to transtuzumab and/or anthracyclines	II	Ongoing	Capecitabine, Lapatinib	[158]
	mAb: MK-0646	ER+/HER2-BC TNBC	II	Completed	Ridaforolimus, exemestane	[159]
IGF-1R RTKI	OSI-906	Hormone-sensitive metastatic BC	II	Terminated	Erlotinib, Letrozole, Goserelin	[160]
	BMS-754807	HER2+ advanced or metastatic BC HR+, HER2−, resistant to non-steroidal aromatase inhibitors	II	Completed	Trastuzumab, Letrozole	[161]

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
