# Peer review of "Decoding the Role of Insulin-like Growth Factor 1 and Its Isoforms in Breast Cancer"

_ijms, 2024, doi:10.3390/ijms25179302_

Round 1
Reviewer 1 Report
Comments and Suggestions for Authors
The article " Decoding the Role of Insulin-like Growth Factor-1 and Its Isoforms in Breast Cancer" by Amalia Kotsifaki et al. provides valuable insights. However, there are several areas that require attention:
1. The authors should clearly state the aim of the study and conclusion in the abstract.
2. The authors should clarify the number of years reviewed in the indicated database and the keywords used for bibliographic research in the introduction.
3. The authors can provide a comprehensive table containing in vitro and in vivo evidence of the role of IGF-1 in breast cancer proliferation, migration, invasion, and metastasis using breast cancer cell lines and xenograft breast cancer animal models.
4. The authors can provide more evidence to support their claims and discuss the limitations of existing research in this area.
5. The authors mentioned the potential of targeted therapeutic strategies against the IGF-1 signaling pathway in BC treatment without providing any concrete discussion of the challenges involved in developing such therapies.
6. The authors can discuss the limitations of existing studies or the potential biases or confounding factors that may influence their results.
Comments on the Quality of English LanguageMinor editing of English language required.
Author Response
Reviewer 1, thank you for your thoughtful and detailed comments on our review. Your insights and feedback are greatly appreciated and have been invaluable in enhancing the quality and clarity of our work. We have carefully considered your suggestions and incorporated the necessary revisions to address your concerns.

Reviewer 2 Report
Comments and Suggestions for Authors
Dear Authors,
Thank you for the opportunity to review your manuscript. I found your work to be highly interesting and relevant, particularly given the current focus on the role of IGF-1 in breast cancer.
Overall, the manuscript appears to be well-structured; however, in my opinion, the introduction and discussion sections are the weaker parts of the paper and would benefit from some improvements.
Introduction: In the introduction, the epidemiological data on breast cancer, the information on the classification of breast cancer (histopathological profiles), and the details on Insulin-like Growth Factor 1 should be condensed, as these topics are elaborated upon in subsequent sections. This would help streamline the introduction and allow for a more focused presentation of the main objectives of the review. Furthermore, I strongly recommend clarifying the scope and objective. Consider revising the introduction to clearly state the main research questions or hypotheses that the review aims to address. Additionally, the purpose of the manuscript should be expressed more concisely. This will help guide the reader and set clear expectations for the content that follows.
Although the introduction addresses the relevance of the IGF-1 system in breast cancer, it would benefit from offering more context about the existing gaps in the literature and the reasons why this review is especially timely and necessary. Highlighting the clinical implications of IGF-1 research and how this review contributes to ongoing discussions in the field could enhance the introduction's impact.
Discussion: The discussion briefly mentions the potential clinical implications of IGF-1 in breast cancer treatment, but this section could be expanded. Consider discussing in more detail how the insights gained from this review might influence future therapeutic strategies or impact patient management.
Minor Comments:
- There are a few typographical and grammatical errors throughout the manuscript. A thorough proofreading would be beneficial.
- Consider providing a short section on the potential limitations of narrative reviews, such as the possibility of selection bias and the lack of a systematic approach to literature inclusion.
- I recommend selecting additional keywords that are not already included in the title of the manuscript. This will help increase the visibility and discoverability of your work in relevant databases and search engines.
I hope my comments and suggestions will help strengthen these sections and enhance the overall quality of your work.
I appreciate the effort that went into preparing this review, and I hope my comments and suggestions will help strengthen it further.
Thank you
Comments on the Quality of English LanguageThe English language in the manuscript is generally good, but a review for grammar and clarity would be beneficial to ensure the text is as clear and polished as possible.
Author Response
Reviewer 2, we want to express our sincere gratitude for your constructive feedback on our review. Your thorough analysis and suggestions have significantly contributed to refining our manuscript. We have carefully addressed your comments and made the necessary revisions to enhance the clarity and rigor of our work. Thank you for your time and valuable input. Your efforts are greatly appreciated.

Round 2
Reviewer 1 Report
Comments and Suggestions for Authors
Accept in present form
Comments on the Quality of English LanguageMinor editing of English language required.
Author Response
Thank you for your valuable feedback and for highlighting the need to improve the English and correct grammatical errors in the manuscript.
I have carefully revised the entire manuscript, focusing on enhancing the language, correcting grammatical errors, and refining sentence structures to improve clarity. These changes have been implemented to ensure that the manuscript is both scientifically sound and easy to read. However, If you notice any additional errors or areas that still need improvement, please feel free to point out the exact lines, and I will address them promptly.
Thank you once again for your constructive feedback. Your input has been crucial in improving the overall quality of the manuscript.

Reviewer 2 Report
Comments and Suggestions for Authors
Dear Authors,
Thank you for your detailed responses and for addressing all of my concerns. I appreciate the effort you have put into revising the manuscript, and I can see that it has indeed improved compared to the previous version.
I do have one remaining concern regarding the captions for the tables and figures. Have you considered shortening the captions to make them more concise and moving the additional details into the main text? This might help to streamline the presentation and improve the overall readability of your manuscript.
Thank you once again for your work.
Author Response
Thank you very much for your valuable feedback and for recognizing the improvements made to the manuscript. I appreciate your thorough review and your suggestion regarding the captions for the tables and figures.
I have carefully considered your recommendation to shorten the captions and move additional details into the main text. In response, I have revised the captions to make them more concise while ensuring that the essential information is still clearly conveyed. The additional details that were originally in the captions have now been integrated into the relevant sections of the main text. These changes have been highlighted in the manuscript with red color for your convenience.
Thank you once again for your insightful suggestions, which have undoubtedly contributed to enhancing the clarity and readability of the manuscript.
